# AI Derivation and Exploration of Antibiotic Class Spaces

## Abstract

This paper presents a novel machine learning-based approach to fragment-based antibiotic drug design. We introduce a tool called FILTER, which uses chemical structure data, pathway information, and protein targets to predict pharmacokinetic properties of existing and novel drugs. We report on three distinct experiments using FILTER. The first experiment is an in-silico analysis that recreates the historical discovery of penicillin derivatives, validating our approach against known outcomes. The second experiment explores the combination of functional groups from different antibiotic classes to create molecules with multiple mechanisms of action. We refer to this approach as *hybridization* as all synthesized molecules are composed of fragments from both classes. Our final experiment is forward-looking as it explores new chemical spaces to build a library of promising compounds for further antibiotic development. In each of these experiments, FILTER serves as an oracle, predicting physical properties and potential therapeutic efficacy of the new molecular architectures, aiming to accelerate the drug development process and address the challenge of antibiotic resistance. Our approach represents an ongoing, significant shift from traditional drug discovery methods, emphasizing the role of innovative technologies in combating the urgent global threat of antimicrobial resistance.

**Keywords:** fragment-based drug design, antibiotic resistance, pharmacokinetics, hybrid antibiotics, in silico analysis, retrosynthesis, chemical space, machine learning, antibiotic discovery, protein targets

## 1 Introduction

Since the discovery of penicillin, antibiotics have become a cornerstone of modern medicine. However, the relentless emergence of antibiotic resistance has significantly undermined their efficacy, presenting a formidable challenge in the treatment of bacterial infections. Traditional drug discovery pathways, characterized by long timelines and high costs, are proving inadequate in the face of the rapid evolution of resistant bacterial strains. Innovative technologies are imperative to accelerate the discovery of new antibiotics. By integrating machine learning (ML) and artificial intelligence (AI) into chemical synthesis and drug discovery, researchers can leverage computational power to uncover novel compounds and predict their effectiveness, thereby streamlining the development pipeline and reducing both time and expense.

This paper presents a tool, FILTER, and proposes experimental methodologies to explore the antibiotic space in search of antibiotic compounds. By "antibiotic space," we refer to the vast chemical landscape that emerges from the combination and modification of molecular fragments derived from existing antibiotics. This space includes both well-established and unexplored derivatives, offering the potential to uncover new compounds with enhanced antibacterial properties. Borrowing from a concept in organic chemistry involving the deconstruction of known chemical structures into simpler precursor components, our first experiment employs a historical *retrosynthetic analysis* revisiting the evolution of the discovery of antibiotics like penicillin. By taking early precursors within the penicillin class, we recreate and extend the historical trajectory of the development of the penicillin antibiotic class and its derivatives. This retrosynthesis validates our methodology but also sets a precedent for its application to other classes of antibioticss. Such experiments can lead to a rapid

expansion of the known chemical space and the creation of further modern antibiotics and derivatives.

Furthermore, this leads to a second experiment in which we call *hybrid antibiotic design*. The hybrid antibiotic design uses the insights gained from the historical analysis of antibiotic modifications and the predictive capacity of FILTER to guide the synthesis of hybrid molecules combining multiple mechanisms of action. This strategy aims to produce treatments that are effective against a broad spectrum of bacterial pathogens, including those resistant to current therapies. Hybridization involves combining functional groups from multiple antibiotic classes into single molecules. These hybrids will be specifically designed to incorporate multiple mechanisms of action, potentially leading to more effective broad-spectrum treatments against a range of bacterial pathogens, including those resistant to existing therapies.

The ability to generate viable antibiotic candidates from historical data means that the vast records of antibiotic development are no longer just a repository of information but a dynamic toolkit for innovation. This can significantly expedite the drug development process, reducing the timeline from concept to clinical application, which is crucial in addressing the urgent global challenge of antibiotic resistance.

Our final experiment is structured similarly to our second experiment with one major difference: the antibiotics of interest. In this experiment, we diverge from a historical analysis by choosing newer antibiotic compounds and exploring the resulting fragment-based space of molecules.

FILTER and the dynamic framework described in this paper not only enhances our understanding of antibiotic evolution but also drives the innovation of new compounds that can be fast-tracked into clinical testing. Our strategy represents a shift from traditional discovery methods to a more integrated, technology-driven approach that accelerates the development of vital new antibiotics to combat the growing threat of antimicrobial resistance.

The code and the datasets for this paper are available at https://anonymous.4open.science/r/FILTER/.

## 2 REPRESENTATION LEARNING WITH OUR CHEMICAL ORACLE: FILTER

A pivotal tool in our study is FILTER, an AI tool designed to predict the physical properties and therapeutic efficacy of new molecular architectures. FILTER leverages comprehensive chemical structure data, pathway information, and protein targets to identify potential pharmacokinetic properties and interactions within biological systems. FILTER employs predictive modeling techniques that focus on the anticipated protein and protein pathway targets of the synthesized molecules. By integrating data from various biological databases, FILTER can anticipate how new compounds will interact with specific biological pathways, providing insights into their potential efficacy and safety profiles. Additionally, FILTER distinguishes itself from other 'oracle' software Alhossary et al. (2015) by incorporating a docking-based oracle which allows for direct analysis of a generated antibiotic to bind to known targets within its expected antibiotic class Li et al. (2019). FILTER is essential for extending analyses of under-explored antibiotic classes Centers for Disease Control and Prevention (2022). It aids in the prediction of novel compounds that may exhibit distinct mechanisms of action compared to current clinical antibiotics, thereby addressing the growing issue of antibiotic resistance. By identifying molecules that interact with novel targets or utilize different biological pathways, FILTER enhances the likelihood of discovering effective treatments against resistant bacterial strains.

### 2.1 MECHANISMS OF ACTION

A *mechanism of action* (MoA) (or 'mode of action') Parker et al. (2024) refers to the specific biochemical interaction through which a drug substance produces its pharmacological effect. In the context of antibiotics, the MoA typically involves disrupting essential bacterial processes, such as cell wall synthesis, protein synthesis, or DNA replication . Understanding the MoA is crucial for determining both the efficacy of a compound and its potential to overcome existing bacterial resistant mechanisms.

FILTER predicts whether a synthesized molecule will engage its target in a manner that disrupts key bacterial functions, thereby defining its MoA. This analysis is pivotal in determining if the compound

will exhibit therapeutic effects similar to existing antibiotics or if it can introduce novel mechanisms that circumvent current bacterial resistance strategies Sun & Chen (2024). By predicting MoAs that involve novel targets or alternative biological pathways, FILTER supports the development of antibiotics capable of overcoming resistant bacterial strains.

Our methodology addresses several limitations in current approaches to antibiotic discovery: data scarcity, model scalability, and synthesis efficiency. By leveraging historical antibiotic data and employing semi-supervised learning, we compensate for the lack of labeled data often encountered in early-stage drug discovery. The architecture of FILTER allows for efficient processing of large molecular datasets, enabling rapid screening of extensive chemical libraries. The integration of AI-driven retrosynthesis with predictive analytics streamlines the process of identifying and evaluating potential antibiotic candidates.

FILTER is also a docking-based oracle since it further refines MoA predictions by virtually simulating the binding interactions between molecules and target proteins. This simulation ensures that the predicted MoA aligns with the molecule's ability to physically interact with protein targets, thereby enhancing the accuracy and reliability of the MoA predictions. By confirming the feasibility of molecular binding, the docking analysis provides a critical layer of validation for the predicted therapeutic actions of the compounds.

## 2.2 DATASETS

Our study utilizes a comprehensive set of datasets to train and validate the predictive models within FILTER. These datasets encompass a wide range of chemical, biological, and structural information essential for accurate property prediction and determination of MoAs.

- **DrugBank** Wishart et al. (2024). This database provides experimentally-derived physical properties and SMILES (Simplified Molecular Input Line Entry System) representations of molecules. DrugBank serves as a foundational dataset for training models on drug-likeness and pharmacokinetic property predictions.
- **Reactome** Jassal et al. (2020). Reactome offers detailed information on protein-correlated pathways and their associated biological processes. This dataset is instrumental in mapping the interactions between synthesized molecules and biological pathways, aiding in MoA prediction.
- **Protein Data Bank (PDB)** Berman et al. (2000). PDB contains high-resolution 3D structures of protein targets, which are essential for the docking-based oracle component of FILTER.
- **ANTIV Siamese Network Embeddings** Redaction (YEARb). The ANTIV Siamese Network (SNet) model was trained using ASCII string representation that describes the structure of an input molecule representation (i.e., SMILES: Simplified Molecular Input Line Entry System Weininger (1988)). The output of the SNet are embeddings that capture the structural and functional similarities between molecules in the form of a high-dimensional vector representing molecular structures thus facilitating efficient comparison and clustering. In this way, we are transferring the learning from the SNet (which considers context of protein-protein interactions, pathways, and more) to a model that learn from input SMILES. By leveraging these embeddings, FILTER enhances its predictive capabilities, allowing for more accurate assessments of drug-likeness, pharmacokinetic properties, and MoAs.

## 2.3 PREDICTION MODELS AND FEATURES

***Prediction Models and properties predicted by FILTER***. FILTER employs a suite of predictive models to evaluate and predict various physical properties and biological interactions of synthesized molecules. The integration of multiple models ensures a robust and comprehensive analysis of each compound's potential properties and efficacy. The prediction models used within FILTER are listed and described in Table 1. Given physical properties predicted by our models, we then assess the drug-likeness and pharmacokinetic profiles of synthesized molecules; these are enumerated in Table 2 in Section A.1 of the Appendix.

***Select features and their relevance***. While FILTER calculates a broad spectrum of physical properties, certain features are particularly influential in determining a molecule's drug-likeness and therapeutic potential. Some of the most salient features are considered below.

Table 1: Prediction models employed in FILTER.

| Component | Description |
|---|---|
| **Neural Network (NN) Predictions** | Use SMILES representation of molecules to predict a range of physical properties. NNs are adept at capturing complex, non-linear relationships within the data, making them suitable for accurate property prediction based on the structure. |
| **XGBoost Predictions** | This XGBoost Chen & Guestrin (2016) gradient boosting framework predict physical properties using the last layer of the neural network. XGBoost enhances prediction accuracy by effectively handling feature interactions and preventing overfitting. |
| **Predicted Embeddings** | Built around the SNet model, this component generates embeddings from input SMILES, capturing essential chemical features in a high-dimensional space. These embeddings facilitate downstream clustering and pathway analysis. |
| **SNet Embedding Clustering** | Using HDBScan Campello et al. (2013), we cluster similar embeddings to identify protein pathways associated with near-neighbor molecules. This clustering aids in predicting potential biological interactions and MoAs using the Reactome dataset. |
| **Quick Vina 2 (Autodock) Analysis** | Integrates input SMILES and protein targets to output docking scores. This analysis simulates the physical binding of molecules to target proteins, measuring binding affinity Copeland (2000) and specificity through a docking score. |

- **Rule of Five** Lipinski et al. (1997): This heuristic evaluates drug-likeness by assessing molecular weight, lipophilicity (logP), and hydrogen bonding capabilities. Compounds adhering to the Rule of Five are more likely to exhibit favorable absorption and permeation characteristics.
- **Polar Surface Area (PSA)**: PSA is pivotal in predicting a molecule's ability to permeate cell membranes and its overall bioavailability. Molecules with lower PSA values typically exhibit better membrane permeability.
- **Bioavailability**: Measures the proportion of a drug that enters systemic circulation, providing insight into its potential efficacy. High bioavailability is desirable for effective therapeutic action.
- **Rotatable Bond Count**: Influences the molecule's flexibility, which can affect binding affinity and specificity to target proteins. A balanced number of rotatable bonds ensures sufficient flexibility without compromising binding stability.
- **caco2 Permeability** Kus et al. (2023): Predicts the molecule's ability to cross the intestinal epithelium, a critical factor for oral bioavailability. High permeability suggests efficient absorption and systemic distribution.

## 2.4 FILTER AS A REPRESENTATION LEARNING TOOL

FILTER plays a central role in our representation learning framework by providing high-quality, predicted features that enhance the ability of the model to make accurate predictions about novel compounds. FILTER facilitates the incorporation of domain-specific knowledge, bridging the gap between chemical structure and biological activity. Our approach aims to develop a methodology and a library of lead chemicals that pharmacologists can utilize to facilitate their research and discovery of novel therapeutic agents. This synergy between FILTER and our learning models facilitates the discovery of promising antibiotic candidates with optimized properties and novel mechanisms of action. FILTER works in tandem with GEN Redaction (YEARa), our tool for synthesizing compounds. While GEN generates potential molecular structures based on retrosynthetic analysis, FILTER evaluates these structures for their drug-like properties and potential efficacy. This integration allows for rapid iteration of molecule generation and evaluation, significantly accelerating the drug discovery process.

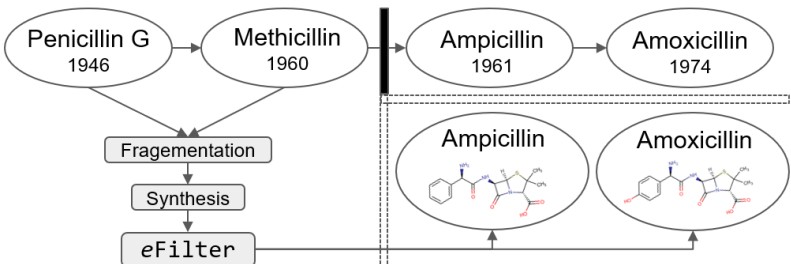

Figure 1: Retrosynthetic analysis of the penicillin class: modeling historical discoveries of derivatives ampicillin and amoxicillin using Penicillin G and Methicillin.

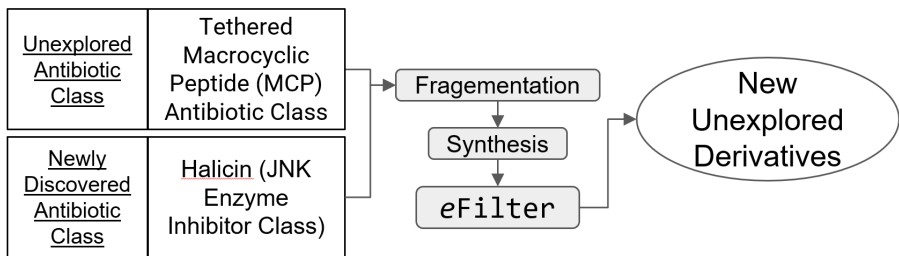

Figure 2: Chemical space exploration of new or unexplored antibiotic classes.

## 3 EXPERIMENTAL METHODS

In this section we consider three experiments using FILTER as a computational oracle, each aimed at furthering our understanding of antibiotic development through an in silico approach.

### 3.1 EXPERIMENT 1: RETROSYNTHETIC ANALYSIS OF PENICILLIN DERIVATIVES

As depicted in Figure 1, the first experiment focuses on the retrosynthetic analysis of the penicillin class with the aim of reconstructing the development history of penicillin derivatives. This approach demonstrates the methodical advancement of antibiotic design and validates the effectiveness of retrosynthetic techniques. GEN is used for to reconstruct more advanced derivatives from simpler antecedents. The output library of compounds is then processed by FILTER to evaluate efficacy and predict potential protein interactions. By cross-referencing AI-generated synthetic pathways with documented historical synthesis routes, we assess the predictive success of the tool.

### 3.2 EXPERIMENT 2: HYBRIDIZATION OF FUNCTIONAL GROUPS FROM MULTIPLE ANTIBIOTIC CLASSES

The second experiment focuses on the design of hybrid antibiotics by strategically combining functional groups from different classes of antibiotics with the aim of creating compounds with multiple mechanisms of action. We selected functional groups on the based on their known efficacy and mechanism of action from the chemical space exploration phase as well as historical domain knowledge. Each synthesized hybrid molecule is analyzed using FILTER to rank leads based on predicted efficacy, expected spectrum of activity (i.e., pathway involvement), and potential resistance evasion capabilities derived from hybridization. In silico docking is conducted to evaluate the interactions of these hybrids with various bacterial targets, further evaluating their broad-spectrum activity and efficacy against resistant strains. The ultimate goal of this approach is to develop a comprehensive library of hybrid antibiotics. This library may contain compounds that not only demonstrate enhanced efficacy, but also incorporate novel or expanded mechanisms of action, thus offering promising candidates for further development.

### 3.3 EXPERIMENT 3: CHEMICAL SPACE EXPLORATION: PATHWAY ANALYSIS

The third experiment involves exploring new chemical spaces by analyzing the biochemical pathways affected and predicted effects of synthesized compounds from novel antibiotic classes. As depicted in Figure 2, our methodology for this experiment is similar to the second experiment. However, in this case, we synthesize compounds from fragments of antibiotic classes in which the derivative space is ill-explored. We again leverage FILTER to assess the potential efficacy of these compounds by simulating their interactions within bacterial metabolic pathways, including docking simulations to predict binding affinities to target proteins. The primary goal of this approach is to build a library of promising compounds, some with novel mechanisms of action and others that expand known mechanisms.

## 4 RESULTS

We present a comprehensive analysis of the FILTER model's performance across multiple tasks and methodologies. We begin by evaluating the model's ability to predict various physical properties of molecules by comparing the performance of neural networks, XGBoost, and a combined approach. Next, we explore the application of Siamese Network embeddings to cluster newly synthesized molecules and to infer their potential biological pathways. Finally, we assess the antibacterial potential of our synthesized compounds through molecular docking simulations.

### 4.1 PHYSICAL PROPERTIES

We evaluated the performance of FILTER across several binary classification and regression tasks for predicting physical properties. These evaluations were conducted using neural networks (NN), XGBoost (XGB), and a combined model that integrates both approaches. A detailed comparison of performance across all target properties is summarized in Table 3 in Section A.2 of the Appendix.

As an example of each task type, we consider bioavailability and PSA predictions. All models exhibited strong performance predicting bioavailability with ROC AUC values above $0.90$. The combined model achieved the highest ROC AUC of $0.9104$, slightly outperforming both individual models. Additionally, it demonstrated high precision ($0.9653$), recall ($0.8385$), and an F1 score of $0.8975$, suggesting reliable performance in predicting bioavailability. For predicting polar surface area, XGBoost again provided the best results with an RMSE of $0.8453$ and MAE of $0.4361$. Although the combined model reduced some prediction error compared to the neural network, XGBoost was the most accurate for this task.

Overall, the combined model consistently provided the best results for binary classification tasks, while XGBoost was the most effective for regression tasks. These findings demonstrate the flexibility and utility of FILTER in predicting both categorical and continuous molecular properties.

### 4.2 CLUSTERING BY SIAMESE NETWORK EMBEDDINGS

Having established the effectiveness of FILTER in predicting physical properties, we next investigated its capacity to capture more complex biological relationships through the use of ANTIV SNet embeddings.

***SNet Model Overview Redaction (YEARb)***. The SNet model utilizes Node2Vec Grover & Leskovec (2016) to generate embeddings for drugs and antiviral peptides (AVPs) from a multigraph of drug-protein and protein-protein interactions. Node2Vec performs random walks through the graph, capturing topological and functional information about each node. The resulting embeddings map drugs and AVPs into a continuous feature space where proximity between vectors represents similarity in biological function.

The SNet is trained using these embeddings to predict the similarity between drug and AVP pairs. The SNet consists of two identical subnetworks that process the drug and AVP embeddings in parallel. During training, the model minimizes a contrastive loss function, encouraging the embeddings of similar drug-AVP pairs to be close together, while pushing dissimilar pairs apart. This enables the model to create meaningful embeddings that reflect the likelihood of a drug sharing antiviral properties with an AVP, such as inhibiting viral entry, fusion, or replication.

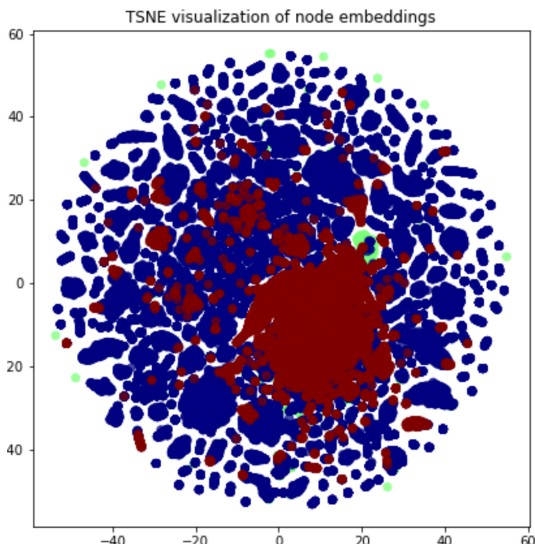

Figure 3: t-SNE plot of SNet embeddings showing clustering of known molecules and newly synthesized molecules; clusters correspond to biological similarity and potential pathway interactions.

***Embedding prediction and pathway clustering***. To predict the pathways for newly synthesized molecules, we trained a model to predict the embeddings of the SNet using only SMILES representations of molecules as input (see Figure 4 in Section A.2 of the Appendix). Importantly, this is knowledge transfer from the SNet–which requires protein-protein interaction (PPI) and pathway data–into a model that can predict similar embeddings based solely on chemical structure.

We then use this model to predict SNet embeddings for our newly synthesized molecules. These predicted embeddings are placed into the same vector space as known drugs, which has pathway information from the Reactome database. By embedding newly synthesized molecules alongside known drugs, we apply the HDBScan clustering algorithm Campello et al. (2013) to group them according to their proximity to drugs with known pathway interactions. This allows us to infer the likely pathways for the new molecules on the basis of their clustering with known compounds.

As depicted in Figure 3, we reduce the dimensionality of the SNet embedding space to visualize this process using t-SNE (t-distributed stochastic neighbor embedding) plot. The plot provides an overview of how newly synthesized molecules are positioned relative to known molecules in the SNet space. Distinct clusters in this space correspond to specific protein pathways and biological functions, offering valuable insight into the functional relevance of these newly synthesized compounds.

By predicting SNet embeddings for new molecules and clustering with known compounds, we are able to assign functional similarities and hypothesize potential protein-pathway interactions, even for molecules without prior biological data. This methodology is especially valuable for prioritizing newly synthesized compounds for further in vitro or in vivo validation.

### 4.3 DOCKING PREDICTIONS WITH QUICKVINA 2

We further evaluate the specific antibacterial potential of our newly synthesized molecules through molecular docking simulations against key antibiotic protein targets. We used QuickVina 2 (qvina02) Alhossary et al. (2015) to perform molecular docking simulations against five E. coli protein targets, specifically penicillin-binding proteins (PBPs), which play a critical role in the efficacy of penicillin-class antibiotics. QuickVina 2 is an advanced docking tool that combines the speed of QuickVina 1 with the accuracy and reliability of AutoDock Vina, making it highly suitable for high-throughput screening of large compound libraries.

QuickVina 2 outputs negative scores, with more negative values indicating stronger binding affinities. To facilitate intuitive comparison, we inverted these scores, making higher values correspond

to stronger binding affinities, which are typically associated with greater antibacterial potential. The primary objective was to identify synthesized molecules most likely to behave similarly to known antibiotics in the penicillin class.

Our focus on PBPs as docking targets is driven by the historical significance and diversity of penicillin as the longest-known antibiotic class. This approach enables us to recreate and evaluate known penicillin derivatives in silico using QuickVina 2 as an oracle for docking predictions, but also test newly synthesized molecules. We ranked the synthesized molecules based on their predicted binding strength to the PBPs. Molecules with docking profiles closely resembling those of known penicillin antibiotics were prioritized as candidates for further experimental validation. This method allows pharmacologists to efficiently narrow down the pool of synthesized compounds, focusing on those with the highest likelihood of exhibiting penicillin-like antibacterial activity. More broadly, the broader goal of the FILTER model is to provide pharmacologists with a framework to evaluate any protein target associated with an antibiotic class of interest, not just penicillin.

The results of our QuickVina 2 simulations revealed a range of binding affinities to our selected PBPs. Top candidates show scores comparable to those of known penicillin-class antibiotics. For example, our highest-scoring novel compound exhibited a binding score of 13.2 compared to ampicillin's score of 10.2 under the same docking conditions. docking shows the predicted binding sites of the proteins we analyzed in our docking simulations.

For the third experiment, we focused on synthesizing molecules similar to the recently discovered antibiotic, Halicin, known for its role as a JNK inhibitor. Using a chemical space exploration approach, we targeted fragments of antibiotic classes that are under-explored. We employed docking simulations to evaluate the interactions of the synthesized compounds. The results, as shown in the Table 4 in Section A.2 of the Appendix, highlight a range of binding affinities, with several compounds exhibiting strong inhibition across multiple JNK proteins. Notably, the top-performing compound showed a binding score of 13.4 against JNK1, closely resembling the binding affinities of Halicin analogs, indicating promising potential for further exploration and optimization.

The results of our QuickVina 2 simulations have significant implications for the field of drug discovery, particularly in the realm of antibacterial drug development. These findings provide a valuable in silico screening method, accelerating the drug discovery process by pinpointing molecules with high potential for antibacterial efficacy. By identifying several promising candidates with strong binding affinities to PBPs, we have demonstrated the potential of FILTER to accelerate early-stage drug discovery.

## 5 RELATED WORKS

Several key studies have laid the foundation for these innovations. One of the contributions in this area is the work by Zhang et al. (2019), which developed a Bayesian semi-supervised graph convolutional neural network (GCN) for predicting molecular properties and improving uncertainty quantification. Although the model showed strong performance in predicting bioactivity, it relied heavily on a large labelled dataset. This approach presents challenges in early-stage drug discovery where labelled data is scarce. Furthermore, while the Bayesian framework enhanced uncertainty estimates, the model's application was limited to molecular structures and lacked integration with chemical retrosynthesis.

Similarly, Schor et al. (2022) introduced the *deepFPlearn* tool, a deep learning-based model designed to predict chemical-gene associations. While deepFPlearn addressed the challenge of predicting chemical effects at a large scale by combining autoencoders and deep feed-forward neural networks (FNN), it suffered from limitations in capturing interactions between more complex molecular architectures. Additionally, the performance of the tool was optimized for toxicology applications and not for antibiotic design, thus limiting its applicability in drug discovery for novel antibiotic compounds.

In recent years, fragment based drug design has been applied with two main strategies for antibiotic discovery: top-down and bottom-up. The top-down approach focuses on repurposing existing molecules, where existing drug-like molecules are incrementally pruned or refined to identify key substructures with antibiotic potential. This method, exemplified by the discovery of Halicin, where AI was used to screen known chemical libraries, leading to the identification of a structurally

unique compounds with novel mechanisms of action, such as disrupting bacterial proton motive force Stokes et al. (2020). This strategy streamlines the search for novel antibiotics by mining existing drug spaces and exploring their potential new applications. In contrast, the bottom-up approach, which our team adopted, focuses on constructing new molecules from smaller fragments derived from known antibiotics. This synthesis-driven method enables the exploration of structurally unique molecules, expanding the chemical space beyond known antibiotics, enabling the construction of structurally unique antibiotics with potentially different mechanisms of action. Using this approach, new classes of antibiotics can be designed and tested in silico before laboratory validation, addressing the need for antibiotics with novel mechanisms to combat resistant strains.

Nicolaou (2014) highlighted advancements in synthetic organic chemistry, particularly in the replication and synthesis of complex bioactive molecules. While this work provides valuable insights into organic synthesis, it focuses primarily on the manual design and synthesis of analogs without leveraging computational tools to accelerate these processes. The study's focus on the synthetic methodology also limited its scope in terms of using AI to predict therapeutic efficacy and physical properties of molecules, a gap we aim to address by integrating AI-based retrosynthesis with predictive analytics.

The MoleculeNet benchmark, introduced by Wu et al. (2018), addresses the need for standardized evaluation of molecular machine learning methods by curating multiple datasets and providing high-quality implementations of molecular featurization and learning algorithms. While this benchmark has enabled significant advancements in molecular property prediction, it still faces challenges in data scarcity and imbalanced classification, particularly for quantum mechanical and biophysical datasets. The use of physics-aware featurizations, such as those leveraging quantum chemistry, has shown promise, but limitations remain in handling more complex molecular architectures and predicting novel compounds.

Our study proposes an AI-driven retrosynthetic approach that not only recreates historical synthesis pathways, such as those used in penicillin production, but also expands them through AI-guided exploration of new chemical spaces. This addresses the data limitations found in Zhang et al.'s Bayesian GCN Zhang et al. (2019), as our semi-supervised learning model leverages historical antibiotic data to compensate for the lack of labeled data. By integrating retrosynthesis with AI tools like FILTER, we also overcome the limitations seen in previous studies by addressing the imbalance problem by focusing on underexplored antibiotic classes and their potential hybrid structures and also by broadening the application of AI to complex antibiotics with hybrid structures. In short, our study bridges these gaps by combining historical retrosynthesis with AI-driven exploration of new antibiotics, addressing limitations in data availability, model scalability, and synthesis efficiency.

# 6 DISCUSSION, FUTURE DIRECTIONS, AND CONCLUSIONS

FILTER offers significant advantages in the context of antibiotic discovery. By integrating neural networks, gradient boosting, and docking simulations, FILTER achieves enhanced predictive accuracy in both property and mechanism of action predictions. This multifaceted approach allows for the comprehensive assessment of a wide array of physical properties and biological interactions, providing a holistic view of each molecule's potential. Moreover, FILTER demonstrates remarkable scalability, efficiently processing large datasets of molecular structures and enabling the rapid screening of extensive chemical libraries. This capability is crucial for accelerating the discovery process and managing the vast chemical space associated with antibiotic compounds.

This approach enables rapid in silico screening of large compound libraries, significantly reducing the time and resources typically required for initial lead compound identification. Moreover, the ability to predict binding affinities to specific protein targets enables a more targeted approach to drug design, potentially increasing the success rate of subsequent experimental phases. Our findings suggest that combining machine learning techniques with molecular docking simulations can bridge the gap between computational prediction and experimental validation, offering a powerful tool for rational drug design. This methodology not only streamlines the discovery of penicillin-like antibiotics but also presents a versatile framework adaptable to other antibiotic targets or protein classes, potentially revolutionizing the drug discovery pipeline across various therapeutic areas.

A particularly noteworthy feature of FILTER is its ability to identify novel compounds with unique mechanisms of action by predicting interactions with previously unexplored protein targets and pathways. This feature is essential for combating antibiotic resistance, as it facilitates the discovery of antibiotics that can overcome existing resistance mechanisms by targeting new biological pathways or utilizing alternative modes of action. Additionally, the integration of structural data from the PDB into docking simulations by FILTER enhances the reliability of MoA predictions by incorporating structural biology insights. This integration ensures that the predicted interactions are not only theoretically plausible but also structurally feasible thereby increasing the likelihood of successful therapeutic outcomes.

While FILTER provides a robust foundation for property prediction and MoA determination, several avenues for future enhancements could further augment its capabilities. Incorporating additional datasets, such as genomic and transcriptomic data, could refine MoA predictions by offering a more comprehensive understanding of biological interactions and resistance mechanisms. Furthermore, implementing more sophisticated docking algorithms and molecular dynamics simulations could improve the accuracy of binding affinity predictions, providing deeper insights into molecular interactions. Developing real-time learning capabilities would enable FILTER to continuously update and refine its predictive models based on ongoing experimental data, ensuring that the tool remains current with the latest scientific advancements. Additionally, creating a user-friendly interface would facilitate easier access to FILTER's predictive insights and allow researchers to customize models according to their specific needs, thereby broadening its applicability and impact in the field of antibiotic discovery.

FILTER serves as a cornerstone in our representation learning framework, effectively bridging the gap between chemical structure and biological function. By accurately predicting physical properties and mechanisms of action, FILTER enables the efficient discovery of novel antibiotics with optimized therapeutic profiles. Its integration of comprehensive datasets, advanced predictive models, and structural analysis tools positions FILTER as an invaluable asset in the ongoing fight against antibiotic resistance.

The application of AI-based techniques, such as those embodied in FILTER, to underexplored antibiotic classes opens new avenues for treatment options, further expanding the arsenal available to combat resistant bacterial infections. The actionable nature of this methodology suggests its potential for broader applications beyond antibiotics, extending to other therapeutic areas where historical compound development can inform future innovations.

In summary, the integration of retrosynthetic analysis and AI not only redefines the boundaries of traditional drug discovery but also sets a new standard for the rapid, efficient, and innovative exploration of therapeutic compounds and chemical spaces. This represents a significant stride toward overcoming some of the most pressing health challenges of our time, offering a promising pathway for the development of next-generation antibiotics capable of addressing the escalating threat of antimicrobial resistance.

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
