# A APPENDIX

## A.1 PHYSICAL PROPERTIES PREDICTED BY FILTER

Table 2: The physical properties predicted by FILTER, along with their descriptions and expected ranges.

| Physical Property | Description | Expected Range |
|---|---|---|
| Rule of Five | A measure of a molecule's drug-likeness based on its molecular weight, lipophilicity, and hydrogen bonding. | 0-5 |
| Ghose Filter | Evaluates drug-likeness considering molecular weight, lipophilicity, and hydrogen bonding. | 0-5 |
| MDDR-Like Rule | Assesses drug-likeness based on molecular weight, lipophilicity, and hydrogen bonding. | 0-5 |
| Rotatable Bond Count | The number of rotatable bonds in a molecule, influencing its flexibility. | 0-20 |
| Bioavailability | Extent to which a molecule is absorbed and available for use by the body. | 0-100% |
| Number of Rings | The number of cyclic structures within a molecule, affecting its stability and binding. | 0-10 |
| H Bond Donor Count | The number of hydrogen bond donors, impacting solubility and binding interactions. | 0-10 |
| H Bond Acceptor Count | The number of hydrogen bond acceptors, influencing solubility and molecular interactions. | 0-10 |
| Physiological Charge | The net charge of a molecule at physiological pH, affecting its solubility and transport. | -10 to 10 |
| Melting Point | The temperature at which a molecule transitions from solid to liquid. | 0-500°C |
| Water Solubility | Degree to which a molecule dissolves in water, affecting bioavailability. | 0-100% |
| Polar Surface Area (PSA) | Total polar surface area of a molecule, influencing membrane permeability and drug transport. | 0-200 Å² |
| Boiling Point | The temperature at which a molecule transitions from liquid to gas. | 0-500°C |
| logP | Measure of a molecule's lipophilicity, affecting its ability to cross cell membranes. | -10 to 10 |
| Refractivity | Measure of a molecule's ability to refract light, related to its volume and electron distribution. | 0-100 |
| pKa (Strongest Acidic) | The pKa value of the most acidic group in a molecule, influencing its ionization state. | 0-14 |
| Polarizability | The ability of a molecule's electron cloud to be distorted by external electric fields. | 0-100 |
| pKa (Strongest Basic) | The pKa value of the most basic group in a molecule, affecting its protonation state. | 0-14 |
| pKa | General pKa value of a molecule, indicating its acid-base properties. | 0-14 |
| logS | Measure of a molecule's solubility in a solvent, impacting its bioavailability. | -10 to 10 |
| Radioactivity | Measure of a molecule's radioactivity, important for safety and regulatory compliance. | 0-100 |
| caco2 Permeability | Assesses a molecule's permeability across the Caco-2 cell line, predicting intestinal absorption. | 0-100% |

## A.2 RESULTS ACROSS ALL MODELS AND TARGET PROPERTIES BY FILTER

Table 3: Summary of results across all models and target properties; results in **bold** indicate the best performance for each metric.

| Task Type | Target Property | Metric | NN | XGB | Combined |
|---|---|---|---|---|---|
| Binary Classification | **Bioavailability** | ROC AUC | 0.9101 | 0.9090 | **0.9104** |
| | | Accuracy | 0.8194 | 0.8367 | **0.8367** |
| | | Precision | **0.9700** | 0.9659 | 0.9653 |
| | | F1 Score | 0.8848 | 0.8974 | **0.8975** |
| Regression | **caco2 Permeability** | RMSE | 1.8883 | **1.8707** | 1.8583 |
| | | MAE | 0.9583 | **0.8674** | 0.8950 |
| Binary Classification | **Ghose Filter** | ROC AUC | 0.8821 | 0.8810 | **0.8828** |
| | | Accuracy | 0.8189 | 0.8166 | **0.8171** |
| | | Precision | 0.8166 | 0.8261 | **0.8246** |
| | | F1 Score | 0.8366 | 0.8314 | **0.8323** |
| Regression | **logS** | RMSE | 2.6052 | **2.3354** | 2.4140 |
| | | MAE | 1.4450 | **1.1462** | 1.2394 |
| Binary Classification | **MDDR-Like Rule** | ROC AUC | **0.9157** | 0.9117 | 0.9135 |
| | | F1 Score | 0.6830 | 0.6922 | **0.6966** |
| Regression | **Polar Surface Area** | RMSE | 0.8869 | **0.8453** | 0.8513 |
| | | MAE | 0.5393 | **0.4361** | 0.4831 |
| Regression | **pKa** | RMSE | 0.5179 | 0.5296 | **0.5184** |
| | | MAE | **0.2335** | 0.2456 | 0.2384 |
| Binary Classification | **Rule of Five** | ROC AUC | 0.8830 | 0.8792 | **0.8817** |
| | | F1 Score | 0.8568 | 0.8606 | **0.8508** |

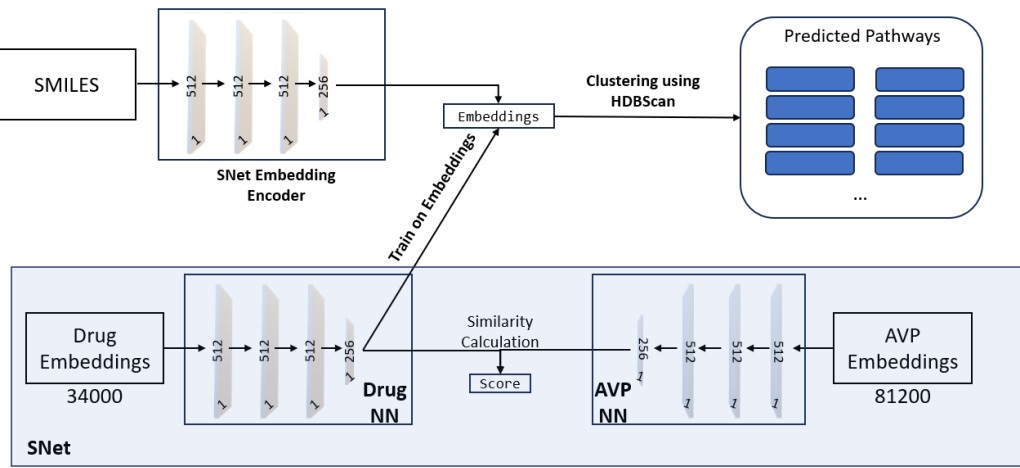

Figure 4: Prediction engine for embedding newly synthesized molecules into SNet space based on chemical structure, enabling pathway clustering and functional analysis by proximity to known drugs with established biological functions in biological space.

Table 4: Top docking scores for synthesized molecules and JNK1, JNK2, JNK3, and FA7 proteins.

| SMILES | JNK1 | JNK2 | JNK3 | FA7 |
|---|---|---|---|---|
| C12Cc3c(c(O)ccc3)C(=O)C1=C(O)C1(O)... | 13.4 | 11.7 | 13.4 | 9.9 |
| C12Cc3c(c(O)ccc3N3CCN(CC3)c3ncc(F)... | 12.6 | 11.0 | 12.6 | 6.3 |
| C12Cc3c(c(O)ccc3n3cc(c4c3cccc4)C3... | 12.4 | 10.8 | 12.4 | 9.4 |
| C12Cc3c(c(O)ccc3n3c(nc(c3)CNC(=O... | 12.4 | 14.7 | 12.4 | 10.2 |
| C12Cc3c(c(O)ccc3n3c(ncc3c3n[nH]c4... | 12.3 | 9.3 | 12.3 | 11.4 |
| C12Cc3c(c(O)ccc3)C(=O)C1=C(O)C1(O... | 10.0 | 14.5 | 10.0 | 9.7 |
| C12Cc3c(c(O)ccc3n3cc(c4c3cccc4)C3... | 9.4 | 13.9 | 9.4 | 7.6 |
| n1(c(ncc1)c1nc[nH]c1c1[nH]cc(n1)C... | 9.8 | 13.9 | 9.8 | 9.0 |
| CCC(=C(c1ccccc1)c1ccc(cc1)ONC(=O... | 9.8 | 10.3 | 13.3 | 7.0 |
| CCC(=C(c1ccccc1)c1ccc(cc1)ONC(=O... | 9.5 | 12.1 | 13.3 | 6.3 |
| CCC(=C(c1ccccc1)c1ccc(cc1)OCCc1n... | 6.5 | 7.3 | 13.3 | 8.0 |