# OpenReview forum: "AI Derivation and Exploration of Antibiotic Class Spaces"
_ICLR.cc/2025/Conference — Submitted to ICLR 2025_

### Official Review · Reviewer_yuRq · 2024-10-23

**Soundness:** 2
**Presentation:** 3
**Contribution:** 2
**Rating:** 3
**Confidence:** 4

**Summary:**

In the paper, the author proposed a novel AI-driven tool for accelerating antibiotic drug discovery. FILTER uses chemical structure data, protein targets, and pathway information to predict properties of both existing and novel antibiotics.

The FILTER model proposed uses neural networks and XGBoost to predict how new compounds will interact with target proteins, and it integrates molecular docking simulations. While the concept is valuable, the paper still have several disadvantages which requires further improvements.

**Strengths:**

1. The model is novel, it merges computational techniques with chemical synthesis for antibiotic discovery.

2. The experiment part is well documented.

**Weaknesses:**

1. Although the FILTER model demonstrates results in silico, future work is needed to validate the predicted compounds experimentally to solidify the claims.

2. The paper requires further effort to show FILTER can be robust when applied to other drug classes.

3. The paper needs more thorough comparison with peer AI-driven drug discovery tools.

**Questions:**

Have any of the compounds predicted by FILTER been validated in laboratory experiments?

---

### Official Review · Reviewer_C3PS · 2024-10-31

**Soundness:** 2
**Presentation:** 1
**Contribution:** 1
**Rating:** 3
**Confidence:** 4

**Summary:**

The authors developed FILTER, a pipeline for antibiotics discovery that uses chemical structure data, pathway information, and protein targets to predict the pharmacokinetic properties of existing and new drugs.

**Strengths:**

no

**Weaknesses:**

This article is not innovative and at the same time does not give me any new insights. The article presents an integrated process for antibiotic screening, but all use existing methods that lack innovation. Moreover, the logic in the paper is confusing, in the introduction it is proposed to be an antibiotic design, but in reality it is a virtual screening of antibiotics. As well, each picture is hard to understand, especially Figure 3.

**Questions:**

1. the logic of the article is confusing, the author proposes 3 experiments, so what is the purpose of these 3 experiments and what is the final result achieved?
2. the article doesn't have a specific flow chart, all the diagrams don't know what they are expressing. Especially in Figure 3, what do the 3 colors represent, and what does this dimensionality reduction diagram illustrate?
3. The method proposed by the author lacks innovation and is based on existing methods for packaging.

---

> ### Comment · Reviewer_C3PS · 2024-11-22
> **There was no response from the author**
>
> There was no response from the author

---

### Official Review · Reviewer_V1B9 · 2024-10-31

**Soundness:** 2
**Presentation:** 1
**Contribution:** 2
**Rating:** 3
**Confidence:** 3

**Summary:**

This paper proposes a fragment-based methodology for antibiotic drug design by integrating multiple machine learning models to predict physical properties, structural data, pathway information, and perform docking simulations for protein targets. The authors focus on filtering target molecules generated through retrosynthetic analysis and selecting functional groups from various antibiotic classes to design antibiotic molecules.

**Strengths:**

The authors outline procedures for designing antibiotics using multiple machine learning models to predict various properties, identify related pathways through embeddings and clustering, and perform docking simulations. These procedures are sound.

**Weaknesses:**

- From the perspective of retrosynthesis and molecular generation with machine learning, the method primarily employs standard ML techniques such as neural networks and XGBoost for property prediction, embedding, clustering methods (HDBScan) for identifying similar pathways, and docking simulations, which lack novelty.
- The experiments are insufficient to demonstrate the efficacy of the proposed methodology. There is no clear explanation of how functional groups are selected, how molecules are generated, or how many molecules are generated and filtered.
- The authors conduct experiments on a specific antibiotic class, but applying the proposed method to other classes is challenging due to its reliance on domain-specific knowledge.

**Questions:**

See the weaknesses

---

### Official Review · Reviewer_9kbH · 2024-11-08

**Soundness:** 2
**Presentation:** 1
**Contribution:** 1
**Rating:** 3
**Confidence:** 5

**Summary:**

This work proposed a machine learning framework called FILTER that sought to predict chemical properties of exisiting and novel antibiotics. By integrating data from multiple sources, FILTER attempted to arrive at novel understanding of the mechanism of action (MoA) of the drugs. Overall, however, the presentation of the work was confusing and a lot of the claims made in the work could not be assessed due to lack of details.

**Strengths:**

- FILTER attempted to add mechanistic insights by analysing of biochemical pathway data.

**Weaknesses:**

- Despite claiming to have performed 3 different set of experiments, only some them (prediction of chemical properties and docking) were shown.
- For predicting chemical properties, FILTER only made use of XGBoost and "neural network", and no details on the architecture of the latter could be found.
- While a work may not necessarily design new model architecture, generally one would expect instead to find applications to novel datasets. Neither aspect could be found in this work. The task-dataset combinations in this work have been studied extensively in the past, with the possible exception of [DrugBank 6.0](https://academic.oup.com/nar/article/52/D1/D1265/7416367) published less than 1 year ago and contained $ 72 \ \\% $ more FDA-approved drugs.
- The properties in Table 3 is only a proper subset of those listed in Table 2. Yet the caption of Table 3 reads "Summary of results across all models and target properties..."
- The presentation in this work was generally quite repetitive and a lot of the contents in methods and results sections can be moved to the discussion section.

**Questions:**

- More details on the datasets:
    - For the chemical property prediction tasks, how did the input/output correspond to the different datasets listed in $ \S $2.2?
    - How have previous works made use of the same datasets (including earlier versions thereof), and what exactly are the novelties in this work?
- While the work made extensive references to "retrosynthesis" and "historical analysis" (cf. $ \S $3.1), how exactly were these done?
- For the docking experiments, how were the "synthesized molecules" (Table 4) similar to halicin generated or selected? Have you tried to generate other molecules based on other existing drugs?
- How were the 3 different aspects of the molecules (chemical properties, molecular pathways and docking affinities) related to each other?
- What do the different colours in Figure 3 represent (presumably different classes of molecules)?
- More generally, could you provide detailed exposition of *all* models trained in this work? This would include information on each layer of neural networks, hyperparameters and training specifications. If helpful, use both tables and figures.
- Could you also provide information on hardware used, training time and validation loss plots?

---

### Meta-Review · Area_Chair_exc7 · 2024-12-19

**Metareview:**

This paper proposes an AI-based tool for predicting pharmacokinetic properties, aiming at effective AI-driven antibiotic drug design.
The reviewers raised multiple major concerns regarding the lack of clarity and need for improving the overall presentation of the work, unclear methodological contribution and technical novelty, and insufficient demonstration of the efficacy of the proposed approach.

**Additional Comments On Reviewer Discussion:**

The authors have not made any effort to address the reviewers' concerns during the rebuttal period and the reviewers doubts and concerns remain unaddressed.

---

### Decision · Program_Chairs · 2025-01-22

Reject